# Quantifying and Cataloguing Unknown Sequences within Human Microbiomes

Sejal Modha,[a] David L. Robertson,[a] Joseph Hughes,[a] Richard J. Orton[a]

ᵃMRC University of Glasgow Centre for Virus Research, Glasgow, United Kingdom

Joseph Hughes and Richard J. Orton contributed equally as last authors. Their order was sorted alphabetically by surname.

**ABSTRACT** Advances in genome sequencing technologies and lower costs have enabled the exploration of a multitude of known and novel environments and microbiomes. This has led to an exponential growth in the raw sequence data that are deposited in online repositories. Metagenomic and metatranscriptomic data sets are typically analysed with regard to a specific biological question. However, it is widely acknowledged that these data sets are comprised of a proportion of sequences that bear no similarity to any currently known biological sequence, and this so-called "dark matter" is often excluded from downstream analyses. In this study, a systematic framework was developed to assemble, identify, and measure the proportion of unknown sequences present in distinct human microbiomes. This framework was applied to 40 distinct studies, comprising 963 samples, and covering 10 different human microbiomes including fecal, oral, lung, skin, and circulatory system microbiomes. We found that while the human microbiome is one of the most extensively studied, on average 2% of assembled sequences have not yet been taxonomically defined. However, this proportion varied extensively among different microbiomes and was as high as 25% for skin and oral microbiomes that have more interactions with the environment. A rate of taxonomic characterization of 1.64% of unknown sequences being characterized per month was calculated from these taxonomically unknown sequences discovered in this study. A cross-study comparison led to the identification of similar unknown sequences in different samples and/or microbiomes. Both our computational framework and the novel unknown sequences produced are publicly available for future cross-referencing. Our approach led to the discovery of several novel viral genomes that bear no similarity to sequences in the public databases. Some of these are widespread as they have been found in different microbiomes and studies. Hence, our study illustrates how the systematic characterization of unknown sequences can help the discovery of novel microbes, and we call on the research community to systematically collate and share the unknown sequences from metagenomic studies to increase the rate at which the unknown sequence space can be classified.

**KEYWORDS** dark matter, genome assembly, human microbiome, metagenomics, microbial dark matter, novel sequences, unknown sequences, virus

Address correspondence to Sejal Modha, s.modha.1@research.gla.ac.uk.

The authors declare no conflict of interest.

Metagenomics has become an increasingly mainstream tool to catalogue the microbial makeup of any given habitat (1–4). It has been applied to a diverse range of environments from human body sites (5–8) to the depths of vast oceans (9–11). Metagenomics, compared to culture-based methods, provides a relatively unbiased approach to observe, measure, and understand the interactions of the microbes within communities as well as with their hosts (3). Underpinned by relatively low sequencing costs and providing powerful insights, metagenomics has become a routine technique to study the microbial content of any environment (2).

These advances in sequencing technologies and the importance of data sharing for reproducible research have led to the rapid expansion of publicly available sequence data. This has led to rapid growth in online sequence databases such as GenBank, which store nucleotide and protein sequence data from various organisms (12, 13). However, although the raw sequences generated as part of metagenomic experiments are made publicly available through the Short Read Archive (SRA) or European Nucleotide Archive (ENA) repositories, the complete set of assembled contigs from a study is rarely submitted to online databases (14). The reason for the absence of this type of data can be associated with the sheer number of contigs generated and the requirement for sequences to be annotated before their submission, which is difficult when the organism the sequence came from is unknown, and when the number of contigs is large. Additionally, taxonomically unidentifiable contigs are typically discarded and excluded from downstream analyses (Fig. 1a), but such sequences represent novel and potentially widespread biological entities, and cataloguing their sequences and where they are found will aid taxonomic classification and our understanding of their biological nature in the future.

The raw data in public databases are typically analyzed using metagenomic protocols designed to address specific biological questions. There is a range of different tools and pipelines available for metagenomic sequence analysis, but there are limited comparisons of these pipelines as they are usually developed to address specific research questions. For example, there are approximately 50 workflows available for virus metagenomic analysis that were used in different publications with primarily different aims (15). As part of the routine metagenomic analysis, only the contigs that can be classified using a specific workflow and that are of interest to the scientific study are typically submitted to sequence repositories such as GenBank. The current approaches used for metagenomics extensively rely on similarity searches to known organisms and proteins; thus, research suffers from the streetlight effect, i.e., observational bias which occurs when people search for something only where it is easier to look. However, in a typical metagenomic data set, a range of assembled contigs cannot be functionally or taxonomically classified, a large proportion of which, even after excluding spurious contigs, bear no functional or sequence similarity to any known sequences and are often referred to as unknown or "dark" sequence matter (16–19). Although the terminology itself has been controversial (19, 20), it typically refers to the sequences of unidentified taxonomic and/or functional origin (Fig. 1b). Generally, these unknown contigs (UCs) are excluded from downstream analyses. However, a number of recent studies have highlighted the importance of identification and categorization of such unknown sequences (17, 19, 21, 22).

Characterization of metagenomically assembled genomes (MAGs) as microbial origin has strengthened the hypothesis that uncharacterized biological sequence matter is highly likely to belong to uncultured or unculturable bacteria, archaea, and viruses present in the microbiome sampled (4, 17, 19, 23). A study by Almeida et al. (24) mined over 11,850 human gut microbiome data sets and identified nearly 2,000 novel uncultured bacterial species from 92,143 genomes assembled from metagenomics data sets. Similarly, another study focusing on multiple human biomes assembled 150,000 microbial genomes from 9,428 metagenomic data sets (25). The MAGs generated from these studies were consolidated to create a unified catalogue of 204,938 gut microbiome reference genomes (26). A range of different data mining studies has led to the identification of novel microbes, including the identification of novel bacterial and archaeal phyla and superphyla (17, 27).

Previous studies have shown that sequences of unknown lineage and unknown functions tend to be of viral origin (16). For example, a computationally identified phage, crAssphage, has been shown to constitute approximately 1.7% of all fecal metagenomic sequences (28). A study by Roux et al. (21) mined 14,977 publicly available bacterial and archaeal genomes and identified 12,498 completely novel viral genomes linked to their hosts. Kowarsky et al. (29) found that 1% of cell-free DNA sequences

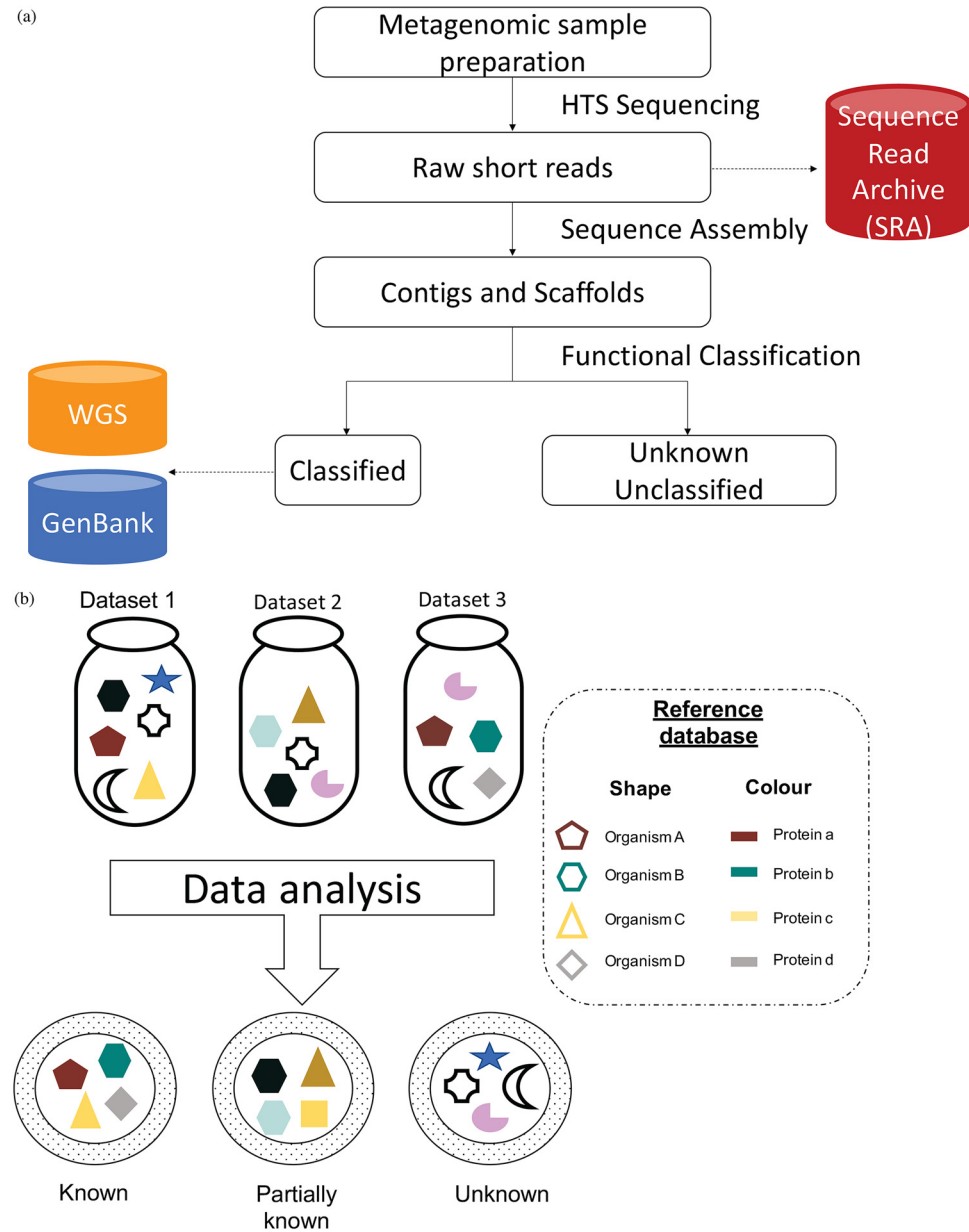

**FIG 1** Typical metagenomic analysis and data submission to public repositories. Overview of existing metagenomic analytical workflow and the definition of unknown sequence matter. (a) Typical metagenomic analytical workflow with data submission steps. (b) A schematic representation of known, partially known, and unknown sequence matter in the metagenomic data sets. HTS, high-throughput sequencing; WGS, whole-genome sequencing.

appear to be of nonhuman origin in human blood samples and only a small fraction of them can be mapped to currently known microbial sequences. Despite this, the characterization of unknown sequences in publicly available data repositories remains an ongoing challenge in microbiome research (4), and the identification of viruses in such UCs remain an even greater challenge due to the absence of universal gene signatures and the high diversity in virus genome content (30). Overall, this highlights the widespread existence of potentially novel viruses and bacteria in the currently available sequence data sets and that a systematic method to identify and catalogue them, especially in human data sets, would be extremely useful. The European Bioinformatic Institute (EBI) has developed MGnify, which enables researchers to analyze their data using a standard metagenomic workflow (31, 32). Similarly, there have been other

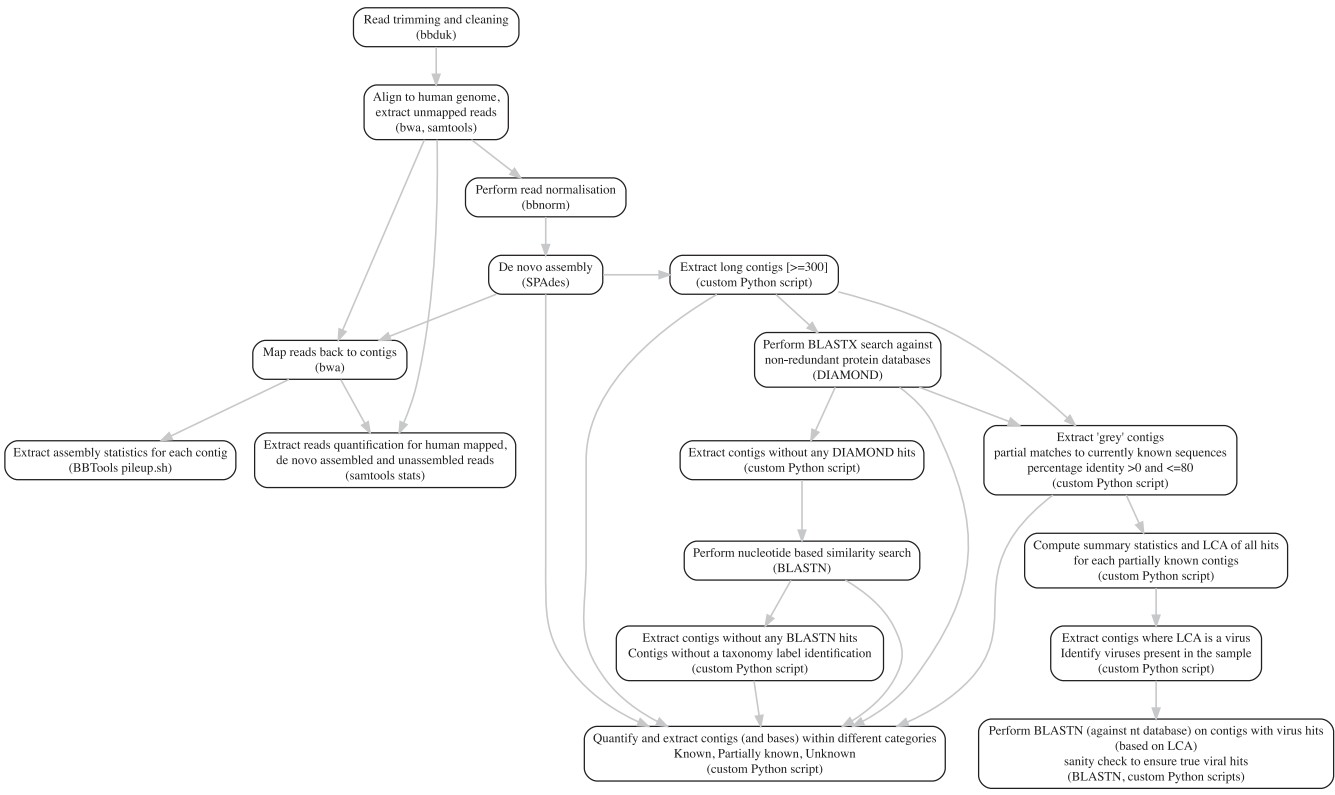

**FIG 2** UnXplore workflow designed to identify unknown sequences in metagenomic data sets. Detailed workflow of the metagenomic analysis and unknown sequence identification pipeline.

community initiatives developed to forward this field of research (31, 33–38). Here, we have focused on the development of a robust, portable, and reproducible analysis framework that aims to identify and quantify the UCs in different microbiome samples (Fig. 2).

In this study, (i) we develop a framework to quantify the unknown sequence matter in human metagenomic data sets; (ii) we compare the unknown sequences between samples, studies, and microbiomes to determine whether these sequences are likely to be of biological origin and whether they are broadly distributed; and (iii) we compare the unknown contigs to currently known sequences in GenBank over the period of the study to determine the rate at which these unknown contig sequences are being taxonomically classified. All unknown sequences and associated metadata have been made publicly available for the research community and the original submitter.

## RESULTS

To quantify the presence of unknown sequences in human metagenomes, data sets included in the EBI MGnify were filtered to select for metagenomic data sets sequenced on the Illumina platform (see Materials and Methods). A set of 963 samples from 40 studies covering 10 different microbiomes were downloaded from SRA repositories and analyzed using the framework described in Materials and Methods in order to characterize and quantify the unknown sequences in these samples. The studies included a total of $2.08 \times 10^{12}$ bases of raw sequence data that were derived from a range of human microbiome studies including the following microbiomes (Fig. 3a): 1, circulatory system ($n = 2$); 2, fecal ($n = 20$); 3, lung ($n = 1$); 4, oral ($n = 5$); 5, pulmonary system ($n = 1$); 6, saliva ($n = 3$); 7, skin ($n = 2$); 8, sputum ($n = 2$); 9, vagina ($n = 1$); and 10, human ($n = 3$; miscellaneous). Geolocation information available for 861 of these samples shows that the data sets are globally distributed but skewed toward western Europe (Fig. 3b and Fig. 4). All samples were individually processed through the metagenomic analysis framework designed in

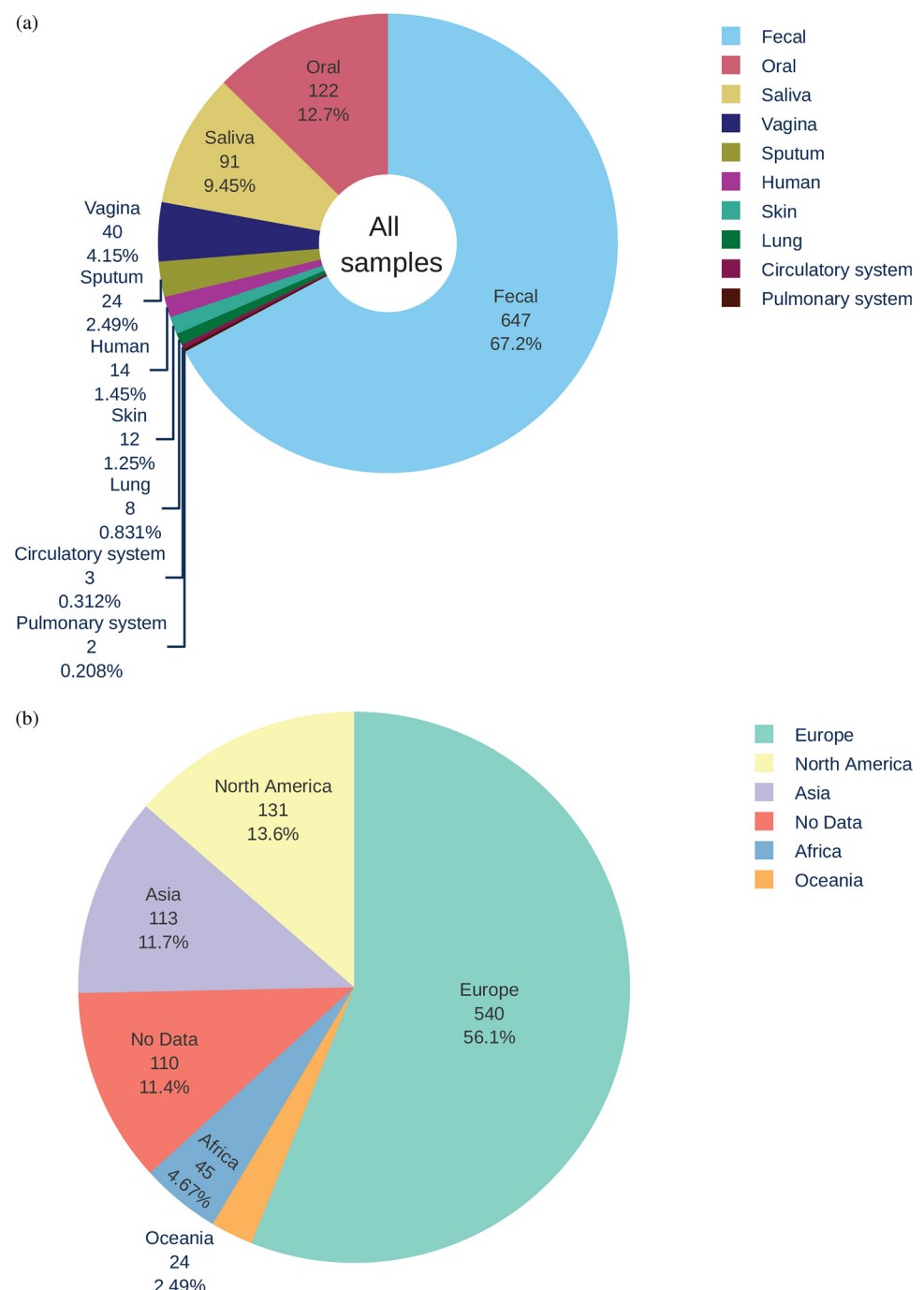

**FIG 3** An overview of the human microbiome data set included in this study. (a) Distribution of samples included in this study for each microbiome (*n* = 963). (b) Overview of the geographical distribution of the samples included in the study (*n* = 861) colored according to the distinct microbiome. The size of the slice represents the number and the proportion of samples. Note that as Russia spans two continents, Asia and Europe, samples from Russia were included in Europe to simplify the illustration in this figure.

this study (see Materials and Methods). The framework included an individual sample-based *de novo* assembly step resulting in a total of 44,238,374 *de novo*-assembled contigs, 28,505,777 of them longer than 300 nucleotides. Out of this set, 7,155,624 contigs were at least 1 kb long, 970,507 were at least 5 kb long, and 415,719 were at least 10 kb long. The largest assembled contig was 1,380,230 bases long and was found in the human gut microbiome sample ERR505090. These contigs were then systematically processed by our metagenomic framework for BLASTX sequence similarity classification against the

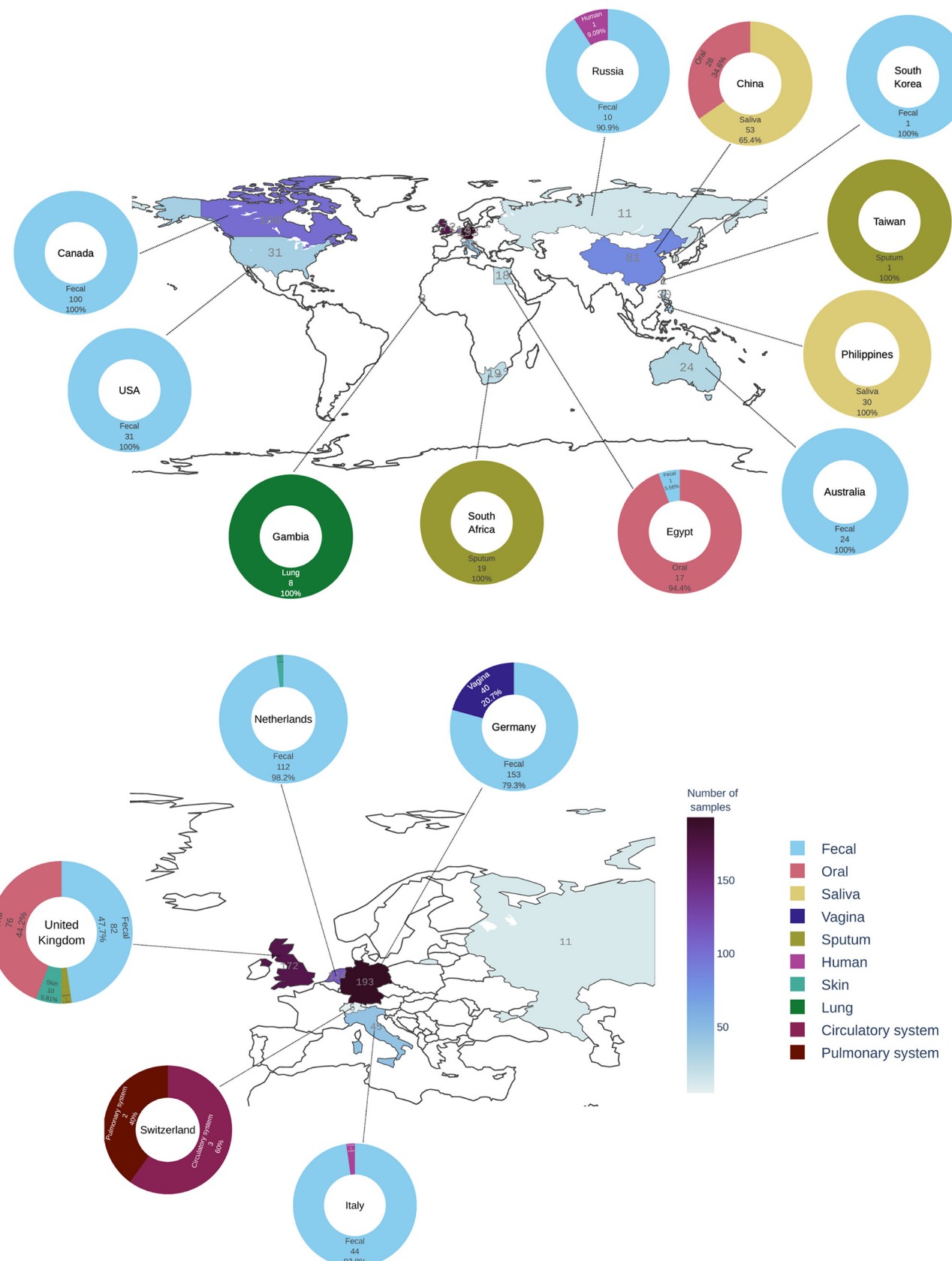

**FIG 4** The geographical distribution of human microbiome samples included in this study. Geographic locations are colored according to the number of samples (*n* = 861) with darker shades representing a higher number of samples analyzed. Samples originating from each location are represented by a doughnut chart. Each doughnut is colored according to the microbiome, and its proportion is represented by the slice of the doughnut.

GenBank nonredundant protein database. Sequence similarity thresholds were used to sort the contigs into three classes: known (>80% similarity to a known protein sequence), partially known (>0 and <80% similarity to a known protein sequence), and unknown (no similarity to any existing sequence).

In total, 25,148,829 (88.22%) contigs were classified as known contigs while 2,517,700 (8.83%) of all analyzed contigs were classified as partially known. The remaining sequences, referred to as unknown contigs (UC), are sequences that did not bear significant similarity to known sequences in the databases. Overall, 651,529 (2.29%) contigs did not match any currently known sequences using our approach and were categorized as UCs. On average, 1.3% of assembled bases per sample were found to be unknown. The proportion of unknown varied significantly between different assembled metagenomes as shown in Fig. 5a. Samples from some microbiomes such as the circulatory system did not contain any unknown sequences, in contrast to the skin microbiome, where this proportion was up to 25.85% for some samples.

The UCs varied largely in length, and most of the UCs were 300 to 1,000 nucleotides long (Fig. 5b). Of all UCs, 95.36% ($n = 621,302$) were shorter than 1 kb, and 4.59% ($n = 29,879$) of UCs were between 1 and 5 kb long. A set of 320 UCs fell within the 5- to 10-kb length category, and 28 UCs were >10 kb long. The largest UC was 42.3 kb long, and the second largest UC was 21.3 kb long. A complete distribution of UCs across different microbiomes is shown in Fig. S1 in the supplemental material, and it shows that the largest UCs were assembled from fecal, oral, and saliva microbiomes.

To understand the coding potential of the unknown sequences, open reading frames (ORFs) were predicted. A total of 273,590 ORFs that were at least 100 amino acids (aa) in length were generated using the standard genetic code. A threshold of 100 aa was selected, which is similar to that used in the taxonomic classification tool GRAViTy, which demonstrated only a 5 to 10% gene loss at this cutoff for viral sequences (39). These ORFs originated from 215,985 distinct UCs, showing that 33.15% of all UCs contained large ORFs. On average, ORFs were 157 aa long with a standard deviation of 87 amino acid residues. The longest ORF was 6,898 aa. This set also included 2,713 ORFs with lengths of at least 500 aa and 256 that were at least 1,000 aa long.

A detailed protein domain analysis for these ORFs was carried out using the InterProScan (40) protein analysis software. This tool searches the domain and functional signature of amino acid sequences against a range of distinct domain databases including Pfam (41), CDD (42), and SUPERFAMILY (43). A total of 36,354 ORFs originating from 35,760 UCs could be functionally annotated using the InterProScan analyses; this number excludes hits to MobiDBLite and Coils databases as they predict disordered regions and coil structure of predicted ORFs as opposed to the domain signatures. An overview of the number of hits found to various InterProScan databases for each microbiome is shown in Fig. S2a. The highest number of hits was found in MobiDBlite (44), a database that can predict the intrinsic disorder regions in the proteins. Overall, 5.49% of UCs ($n = 35,760$) contained ORFs ($n = 36,354$) with at least one identifiable domain. The functional classification of the ORFs was prominently centered around the Pfam database resource (41). Pfam databases facilitate domain-based searches against the set of protein sequences using hidden Markov model profiles. These types of searches can identify distantly related protein sequences. A total of 16,839 ORFs originating from 16,705 UCs were found to match at least one Pfam entry, and in total, 27,025 Pfam hits were derived (Fig. S2a) All Pfam entries were collapsed down to their corresponding protein clans (grouping of related protein families) by mapping the Pfam identifiers (IDs) back to their clan membership. Figure S2b shows a heatmap of the top 50 Pfam clans with hits to UC ORFs predicted in different metagenomes. The most abundant hits were identified to clans tetratricopeptide repeat superfamily and leucin-rich repeats. The largest number of hits was found in the fecal microbiome due to the high number of fecal microbiomes included in this study. Additionally, a range of other protein clans including those that represent helix-turn-helix, beta-strands, polymerase, and nuclease proteins was also found in this set. These results illustrate that the UC sequences have known protein

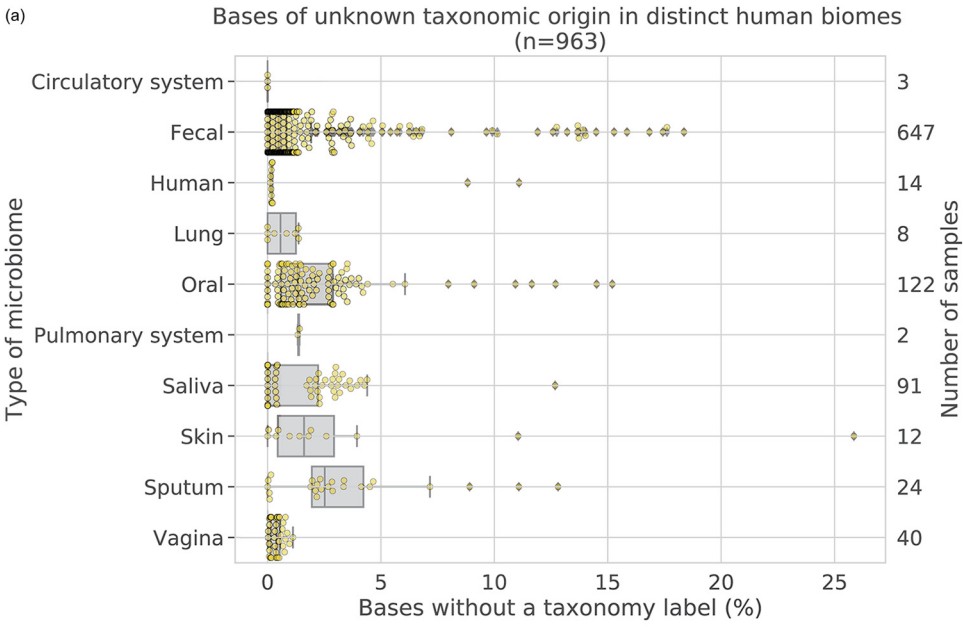

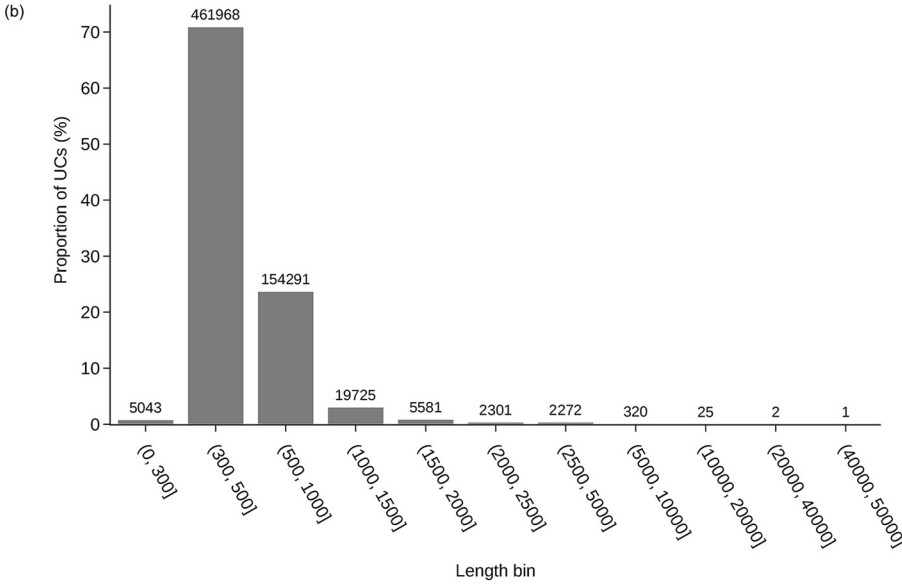

**FIG 5** Quantification of unknown sequences in different human microbiomes. (a) The proportion of unknown bases in different human microbiomes. The proportion of unknown bases was calculated from the unknown contigs for each microbiome. The secondary *y* axis shows the number of samples analyzed in each category. Each individual sample is overlaid on the boxplot and is represented by small yellow circles. (b) The distribution of all unknown contigs in 11 distinct length categories. Each bar represents the proportion of UCs on the *y* axis with the number of contigs in the given category annotated at the top of the bar. Bin sizes are shown in the interval format, which means that sizes are exclusive on start values and inclusive on end values.

domains, suggesting that these unknown sequences are functional and belong to organisms that are not yet fully sequenced or taxonomically classified.

**Unknown sequence clustering.** To investigate the extent of sequence diversity and to identify UC sequences present in multiple samples and microbiomes, sequence clustering was performed. MMSeqs2 (45) generated 464,181 clusters of which 377,855 were singletons, i.e., they did not cluster with any other sequences. These singletons were excluded from the cluster analysis described below. A total of 86,326 clusters comprised two or more sequences with a mean cluster size of 5.7 contigs and a standard deviation of 8.1. Cluster representatives which were the longest sequences in the cluster were extracted from MMSeq's clustering output. The largest cluster contained

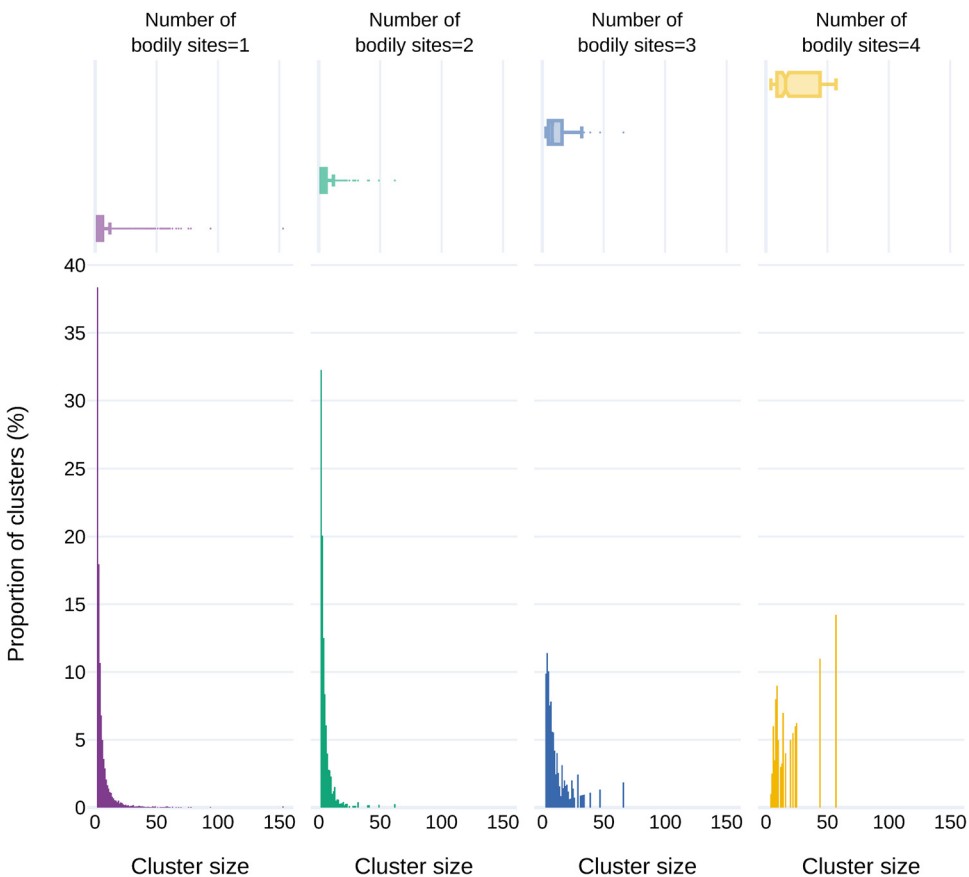

**FIG 6** Distribution of cluster sizes on the *x* axis and their proportion on the *y* axis. The marginal box plot shows the distribution of cluster sizes for each category. The plots are grouped and colored according to the number of distinct bodily sites in which the clusters are found: e.g., number of bodily sites = 2 in green means that members of each cluster are found in data sets from two distinct bodily sites (e.g., gut, skin, fecal, oral), and all clusters from this plot come from 2 distinct bodily sites but may (or may not) come from different bodily sites compared to other clusters within the plot, with one cluster coming from gut and skin, for example, and another from skin and fecal matter, etc.

153 sequences which originated from the fecal microbiome from 8 distinct BioProjects (Fig. S5c). A cluster size distribution across different microbiomes is shown in Fig. 6, and a detailed cluster size distribution with cluster representative length is shown in Fig. S3. Of 273,674 UCs, 89.42% (*n* = 244,730) were clustered into single microbiome clusters, and 10.58% of UCs (*n* = 28,944) were found in clusters that contained sequences from two or more microbiomes. To compare that with specific studies, 39.4% of UCs were clustered into BioProject specific clusters and the remaining 60.6% of UCs (*n* = 165,851) were grouped into clusters originating from two or more BioProjects. A total of 78,139 (90.52%) clusters contained sequences from a single microbiome, and 7,645 (8.86%) clusters included sequences from two microbiomes. Only a few clusters were comprised of members from 3 (*n* = 512) or 4 (*n* = 30) microbiomes. The largest multimicrobiome cluster contained 57 sequences (304 to 9,080 bases long) from 4 distinct microbiomes and BioProjects and contigs assembled from 12 samples. The largest single microbiome cluster contained 153 sequences (6,640 to 300 bases long) from fecal microbiomes with contigs assembled from 46 distinct samples covering 8 different studies. Overall, this clustering method produced very small, study-specific clusters. A set of 464,181 UCs was obtained by combining the cluster representative sequences with the unclustered singleton UCs and used to determine the rate at which UCs are classified.

**Classification of unknown sequences over time.** In this framework, the unknown sequence identification is dependent on the publicly available nucleotide or protein

sequence databases. These data repositories are updated regularly with new sequence data being deposited from around the world. However, typically, the sequence searches are carried out against static versions of the databases. Our analysis conducted against the databases downloaded on 18 April 2019 identified 651,529 UCs that were collapsed down to a set of 464,181 UCs following the cluster analysis. Subsequent analyses on 31 October 2019 and 5 March 2020 produced a set of 613,726 and 558,711 UCs, respectively. The final number of sequences that still lacked a taxonomy label was down to 459,147 after the most recent analysis carried out against the databases downloaded on 14 October 2020. A total of 29.5% ($n = 192,382$) of the sequences compared to the initial set of unknown sequences matched at least one sequence from the updated databases in the BLASTX and the BLASTN steps of the analysis. Similarly, 27.6% ($n = 128,288$) of the representative set sequences could be labeled taxonomically with the updated databases. A rate of taxonomic characterization of 1.64% of unknown sequences being characterized per month was calculated from the complete set. This rate was estimated to be 1.54% for the representative set. Moreover, as shown in Fig. S4, a range of long UCs still remained unknown even after the similarity sequence-based analysis carried out on 14 October 2020.

From a set of 192,382 contigs that were labeled taxonomically after the most recent analyses carried out on 14 October 2020, 167,864 were identified using BLASTX and 24,518 were identified using BLASTN. A total of 106,739 UCs from the BLASTX-classified set were categorized as known, and 61,125 contigs were categorized as partially known. A large majority of these contigs (97.11%, $n = 162,987$) were also deemed to be bacterial. The remaining contigs were divided between cellular organisms ($n = 2,104$), archaea ($n = 930$), viruses ($n = 858$), root ($n = 827$), and *Eukaryota* ($n = 140$). Of all BLASTN hits, 76.55% matched bacteria ($n = 18,768$), 17.88% matched viruses ($n = 4,383$), 1.99% matched *Eukaryota* ($n = 487$), and 0.03% matched archaea ($n = 7$). The hits that could not be mapped to a superkingdom were divided between unidentified plasmid ($n = 544$), root ($n = 294$), cellular organisms ($n = 20$), uncultured organisms ($n = 14$), and synthetic construct ($n = 1$). These results reiterate our initial hypothesis that the majority of UCs represent currently unknown microbial genomes.

**Viral domain signature identification.** One hundred ninety-five UCs were shown to contain a virus-specific functional domain which was parsed using the term "virus" or "viral" in the InterProScan analysis signature description column. Results with the term "phage" were not included in this subset as a range of phage domains is also present in the host bacterial genomes. These domains were predominantly identified using the Pfam ($n = 125$) analysis. The most abundant virus-specific domain was vaccinia virus protein VP39, and it was found in 53 UCs derived from fecal ($n = 23$), saliva ($n = 14$), oral ($n = 12$), sputum ($n = 1$), and human ($n = 3$) microbiomes and was identified by Gene3D analysis. The largest UC containing this domain was 3,661 bases long and was found in sample ERR1474567. Another frequently found domain in the UCs was podovirus DNA encapsidation protein Gp16 domain. It was found in 25 UCs; out of this set, 23 UCs were assembled from fecal microbiome. The largest UC containing this virus-specific domain was a 9-kb-long contig (see Fig. 8a below), assembled from PRJEB18265. This UC was clustered with 24 other sequences (see "Unknown sequence clustering" above) that were assembled from 11 samples representing 5 distinct fecal microbiome studies. These results indicate that these UCs represents a completely novel genome of a virus that is likely related to currently known podoviruses.

The largest UC containing a viral RNA-dependent RNA polymerase (Pfam: PF00680) domain was found in the sputum microbiome sample ERR1022511. This UC was 5,894 bases long and contained seven ORFs that were at least 100 aa long (Fig. 7). A 269-aa-long ORF contained ATPase P4 of the double-stranded RNA (dsRNA) bacteriophage phi-12 (Pfam: PF11602) domain, suggesting that this UC represents the large segment of a novel double-stranded RNA phage which is usually categorized in the virus family *Cystoviridae*. The genomes of these phages are composed of three linear dsRNA segments with a total genome length of 12.7 to 15 kb, and all segments code for various proteins (46). Although several other UCs were found in the same sample, none of

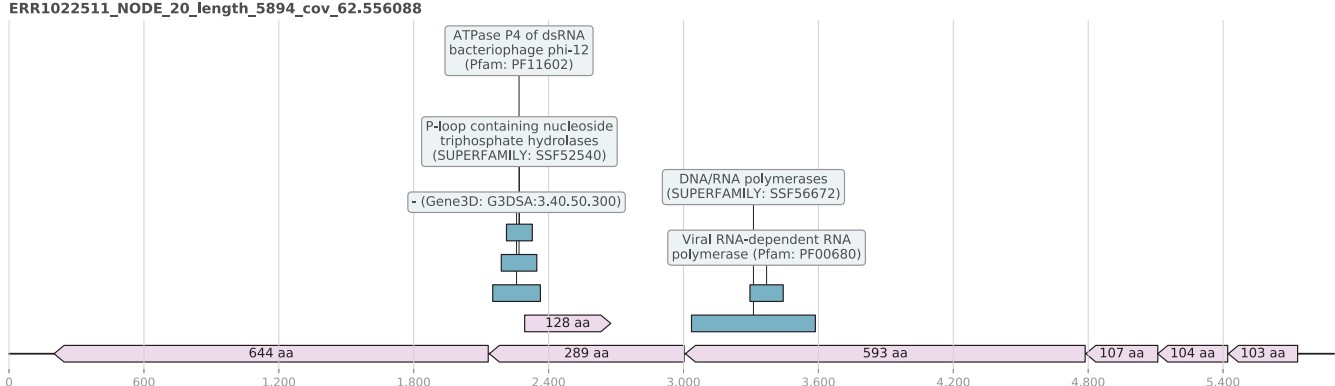

**FIG 7** The genome diagrams of a potentially novel dsRNA phage segment found among the UC set that is hypothesized to be related to currently known cystoviruses. The open reading frames (ORFs) are highlighted in the light pink shade with the ORF lengths as their corresponding labels and the green boxes illustrating the InterProScan-computed presence of domain signature.

them displayed any sequence or functional similarity to the other two segments, i.e., small and medium segments of cystoviruses. However, UCs that could potentially belong to novel cystovirus-like genomes were extracted based on the sequence length, GC content, and sequencing depth criteria. Moreover, this UC representing a potentially novel relative of cystoviruses did not match any known protein or nucleotide sequences even in the most recent analyses, confirming the discovery of a novel virus.

**Virus prediction and comparisons to uncultured virus databases.** From the complete set of the UCs, 323,395 (49.64%) UCs were predicted as viruses by DeepVirFinder (see Fig. S7a). This set included 300,271 UCs that were under 1 kb long, which represent 48.33% of UCs identified in this length category. A number of larger contigs were also predicted as viruses: 76.27% ($n$ = 22,788) of UCs in the 1- to 5-kb length category and 96.55% ($n$ = 336) of UCs in the 5- to 50-kb category. These results strongly support our hypothesis that the large majority of the UCs are of virus origin, albeit a large proportion of short UCs are likely to be fragments of unknown viruses.

These predicted virus sequences ($n$ = 323,395) were clustered with other known and partially known sequences using MMSeqs with 90% sequence similarity across 80% of the sequence. Of UCs, 50.18% (162,271) either were singletons or were clustered with other UCs, while the remaining 49.82% (161,124) of UCs were clustered with known and partially known. However, a large proportion ($n$ = 152,295; 94.52%) of the UCs that clustered with these were shorter than 1 kb. A total of 8,829 UCs (out of 22,788; 38.74%) were at least 1 kb long, among which 1,402 UCs (out of 4,419; 31.73%) were at least 2 kb long, 75 UCs (out of 336; 22.32%) were at least 5 kb long, and 5 UCs (out of 28; 17.86%) were at least 10 kb long. Moreover, 47.52% of sequences that match the UCs were deemed partially known (i.e., had a protein sequence hit with <80% sequence similarity) in this analysis, suggesting that these known and partially known sequences are still significantly divergent from those present in the databases.

To identify the "known unknowns," i.e., uncultured viruses categorized as UCs in this study and also observed in previous meta-analyses, the IMG/VR databases were used as a reference and the UCs were searched against the nucleotide and protein repositories. A total of 182,293 (27.98% of all UCs) UCs had at least one hit to uncultivated viral genomes (UViGs) included in the IMG/VR databases using BLASTN, and 175,372 (26.92%) UCs were found to match at least one UViGs using the BLASTX approach (Fig. S7b). Out of the set of 273,590 predicted ORFs, 85,852 ORFs were found to match protein sequences included in IMG/VR. A total of 64,779 (9.94%) UCs were found to match the uncultured viruses in IMG/VR using all three approaches.

**The large unknown contigs.** All UCs described in this section were predicted to be viruses by DeepVirFinder and did not cluster with known and partially known sequences. The largest UC was assembled from the saliva sample ERR1474583 and was 42,357

bases long. This contig did not cluster with any other contigs and has 23 ORFs that were over 100 aa long. One of the ORFs that is 434 aa long comprised the cysteine proteinase domain (SUPERFAMILY: SSF54001) according to the InterProScan analysis. This contig still remained unknown after searches against the most recent version of the databases, suggesting that the organism to which this genomic sequence belongs is still to be identified and fully sequenced. A snapshot of the ORFs and domain is shown in Fig. 8b, highlighting the presence of coding regions across the entire length of the UC sequence. Based on the results we have obtained here, we predict that this UC sequence is likely to be of microbial origin as it lacks a noncoding region. CheckV analysis predicted it to be a viral genome fragment with the presence of two identifiable viral genes, albeit with low quality per the MIUVIG (47) standards due to the lack of similarity to any known sequences. This strongly suggests that this UC can potentially be a representative or partial genome sequence of a currently unknown and completely novel virus.

A 20,309-nucleotide-long contig from saliva sample ERR1474612 clustered with two very short contigs from the same study. As shown in Fig. 8c, long ORFs were predicted across the whole sequence. Some of the predicted ORFs were found to have interesting domain signatures (Fig. 8c) such as enzymes for nucleic acid replication, e.g., polymerases. An ORF that is 655 aa long shows the presence of the DNA-dependent RNA polymerase domain (SUPERFAMILY: SSF64484). A CheckV (48) analysis of the contig also predicted it to be of viral genomic origin; however, it was predicted to be an incomplete genome. This UC was shown to have a very low identity ($<$30% sequence identity with 2% of query coverage) to a hypothetical protein of a *Firmicutes* bacterium (HAB66316.1) and an AAA-family ATPase from *Sharpea azabuensis* (23% sequence similarity). When the E value threshold was removed, a total of 8 BLAST hits were obtained and 3 out of 8 hits were to a range of phages including *Bacillus* phage vB_BpuM-BpSp, *Vibrio* phage 2 TSL-2019, and *Ralstonia* phage RP12. These hits range from hypothetical to putative proteins. All these matches were localized to a short region between positions 8217 and 8915 which was shown to contain ATPase and P-loop-containing nucleotide triphosphate hydrolase domains (Fig. 8c). Notably, no nucleotide sequence hits were identified for this UC. Although these results have bacterial hits, it is likely that this UC represents a complete or partial genome of a novel phage that infects the host bacteria, e.g., *Firmicutes*.

**Short circular contigs.** A range of circular contigs with direct terminal repeat (DTR) and inverted terminal repeat (ITR) signatures was identified using CheckV in the UC data set. A total of 1,839 UCs containing repeat signatures were predicted of which 1,771 contained DTR signatures and 68 contained ITR signatures. Ninety-four of these UCs were at least 1 kb long, suggesting circular genomes, and 48 of them contained a range of 55-base-long terminal repeats. A cluster of 8 sequences from 2 different microbiomes and studies were identified to contain similar sequences (71 to 100% similarity) assembled from different samples (Table 1). Four cluster members were 2,110 bases long, one sequence was 1,983 nucleotides long, and the cluster representative was 3,165 nucleotides long. The cluster representative sequence contained multiple copies of the same ORFs, suggesting the presence of multiple genome copies, sequencing error, or misassembly. Most of these sequences contained a 50-bp-long DTR sequence signature, GTGCATTTTTTTTGTGCACTTTTTCAAAAAAACCGTGAAAAAAAT TCATT. These contigs contained two distinct ORFs, which were 125 aa and 144 aa long. Similarly, another 50-base-long DTR signature, AATGAATTTTTTTCACGGTTTTTTTGAA AAAGTGCACAAAAAAAATGCAC, was observed in another cluster that had 7 member sequences ranging in similarity from 31 to 100% and assembled from 7 distinct samples. All but one member were 1,770 to 1,771 bases long. These contigs also contained two ORFs that were 102 aa and 106 aa long. These ORFs did not match any existing protein sequences in the databases. These circular contigs were assembled from a range of oral microbiome samples from study PRJNA230363. Similarly, a range of contigs ($n = 9$) that contained inverted terminal repeats (ITRs) were also identified in this

(a)

**ERR1744189_NODE_2461_length_9037_cov_45.156201**

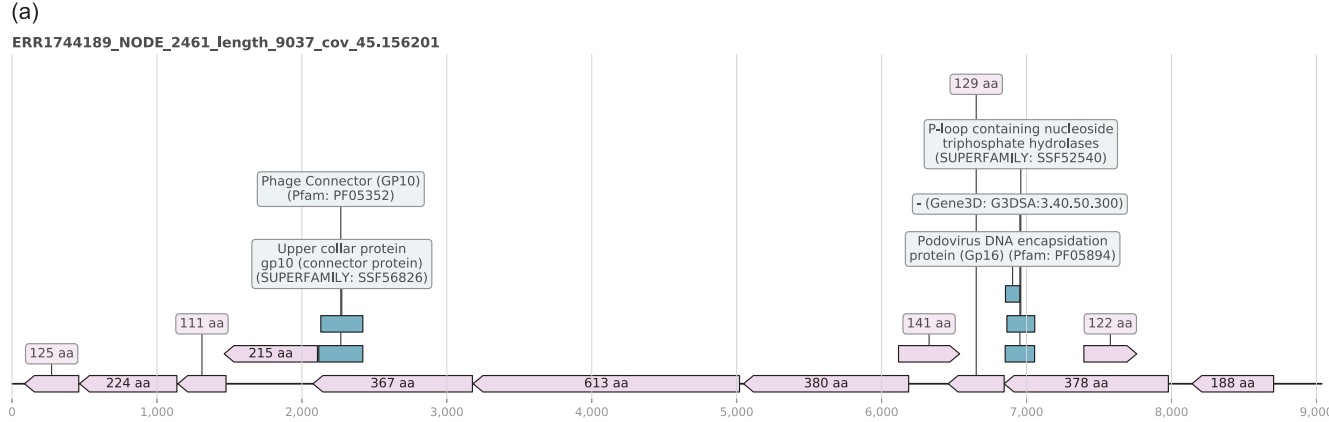

(b)

**ERR1474583_NODE_4_length_42357_cov_4.744622**

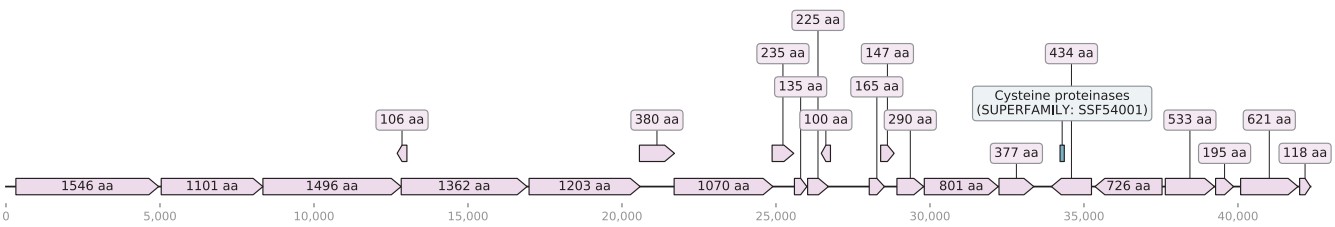

(c)

**ERR1474612_NODE_443_length_20309_cov_34.370495**

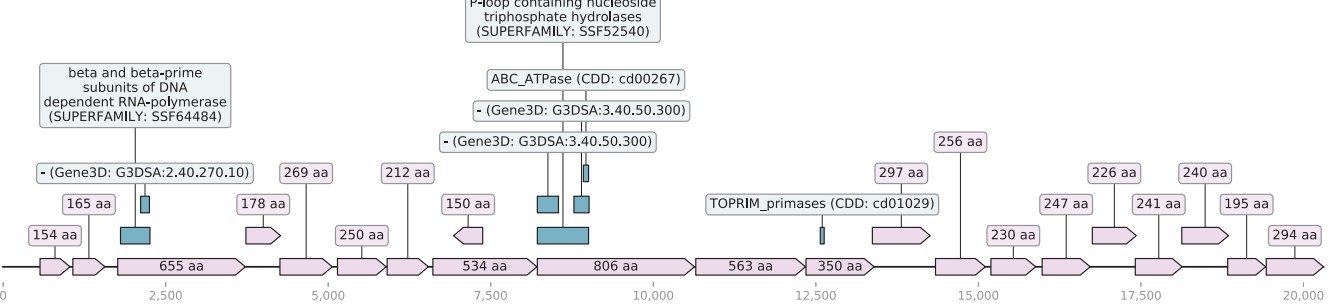

**FIG 8** The genome diagrams of large unknown contigs show the open reading frames (ORFs) in the light pink shade with the ORF lengths as their corresponding labels and the green boxes illustrating the InterProScan-computed presence of domain signature. (a) The largest contig with podovirus DNA encapsidation protein Gp16 domain. (b) The largest unknown contig assembled in the set is categorized as unknown even after the most recent similarity-based search on 14 October 2020. (c) An unknown contig of 20,309 bases in length was described to contain a range of domains including a potential virus-specific RNA polymerase domain.

data set. A cluster of 5 distinct circular contigs was assembled from distinct samples from the fecal microbiome (PRJEB7949). Four out of five of these circular contigs contained the ITR sequence CGAAACGATTGCCCAGAGAGATGACTGTCAATCCGCCCGA TTATTGGGCGCTTAC. They also contained a 138-aa-long ORF. These short circular UCs did not bear any sequence or functional similarity to known sequences or domains, so their biological origin is difficult to predict. However, based on their genome organization and size distribution, we predict that they are likely to represent either novel circular replication-associated protein (Rep)-encoding single-stranded (CRESS) DNA virus groups or novel satellite virus-like groups. Sixteen out of 20 UCs described in Table 1 were predicted to be viruses by DeepVirFinder (see "Virus prediction and comparisons to uncultured virus databases").

**Control samples.** The Human Microbiome Project (HMP) mock community samples (*n* = 9) were downloaded for study PRJNA298489 and were analyzed for quality control

**TABLE 1** Circular contig clusters with direct and inverted terminal repeats

| Study ID(s) | Cluster size (n) | Typical contig length (nt) | Repeat type | Sample type | Sequence similarity (%) (minimum–maximum) |
|---|---|---|---|---|---|
| PRJEB14383; PRJNA230363 | 8 | 2,110 | DTR | Saliva; oral | 71–100 |
| PRJNA230363 | 7 | 1,771 | DTR | Oral | 31–100 |
| PRJEB7949 | 5 | 1,337 | ITR | Fecal | 67–100 |

and workflow assessment. These are control samples that are not expected to yield UCs, but if they did, those UCs could be due to sequencing/assembly error or common lab contaminants. Out of the complete set, four HMP samples did not contain any UCs as expected, whereas SRR2726666, SRR2726669, and SRR2726672 contained one UC each, but their lengths were short, varying from 323 to 449 bases. The remaining two samples, SRR2726670 and SRR2726671, contained 28 and 18 UCs each, respectively. The largest UC assembled in the mock sample was 3,965 bases long and was found in SRR2726670; only 3 UCs were ≥1 kb. These UCs were searched against the most recent version of the databases downloaded on 14 October 2020, and only 8 short contigs—4 from SRR2726670, 2 from SRR2726671, and one each from SRR2726666 and SRR2726669—remained in the UC category. These remaining UCs were only 330 to 513 bases long. These results validate the UC analysis framework developed here and highlight that even in control samples, there is a very minor number of short UCs to be found. New sequence data get uploaded to public repositories daily, and these updated databases contain a greater diversity of sequences, most of which are taxonomically classified. Therefore, UCs identified in the initial analysis of these mock samples were subsequently found to match a known sequence in the updated version of the database as more sequence data were available and classified.

**Resources.** We have developed a modular metagenomic and unknown sequence analysis framework using the sophisticated pipeline management tool Snakemake. Our analysis pipeline takes advantage of portability and flexibility offered by Python, BioPython, and Snakemake tools which allow reproducible analysis of large meta-omic data on any processing servers and clusters. The framework developed here is capable of utilizing multiple cores, enabling users to analyze large data sets in a parallel fashion. This allows the UC data to be properly linked to their original samples and studies. A consistent data labeling scheme is utilized across all studies and samples. For traceability, all UC FASTA identifiers start with SRA sample identifier. All ORFs contain the exact same naming scheme with the suffix "_" and an ORF number starting with 1. A complete metadata table is provided to link any new sequence data to their corresponding BioProject and sample. Functional domain predictions and clustering results are annotated with relevant metadata and provided in a tabular format.

## DISCUSSION

In this study, we have developed an automated framework that can systematically quantify the proportion of unknown contigs (UCs) in meta-omics samples. While the presence of UCs is well recognized, this is the first study that addresses the question of UCs comprehensively and quantifies it across different human microbiomes. Our approach utilizes sequence similarity-based taxonomic categorization to identify the sequences that cannot be categorized. We define these UCs as the sequences that do not match known sequences in the databases with a predefined sequence similarity threshold of E value 0.001, which is a very lenient threshold; anything with an E value higher than this is unlikely to truly be related to the database sequence hit. We show that on average 2.29% of assembled contigs are categorized as unknown in different human microbiome studies. Moreover, a subset of those with unknown sequences could be translated and contained protein domains; thus, we were able to find

functional similarity to 5.49% of taxonomically unknown contigs. We have generated a comprehensive catalogue of 651,529 UCs that do not bear any sequence similarity to sequences present in the widely used GenBank protein and nucleotide databases. Although sequence similarity-based approaches are dependent on the databases, the protein sequence-based approach implemented here is highly effective in fishing out distantly related homologues of known sequences available in the databases (49) and thus provides better resolution for sequence classification than those solely based on the genomic signature-based binning (50). This study highlights the importance of avoiding the "streetlight" effect, i.e., observational bias arising from classifying metagenomic sequences on the basis of related sequences that already exist in the databases. Here, we have aimed to eliminate such observational bias by performing comprehensive data mining of the human microbiome data and cataloguing the UCs, their frequency in different human microbiomes, and their overlap between different samples.

This study has enabled the identification of a range of genomic sequences that are hypothesized to belong to currently uncharacterized organisms that are often found in similar samples and/or microbiomes. A range of large UCs with and without known protein domains is presented here. However, the complete set includes a large number of UCs that still remain unknown and can be mined further to study their biological origin. A third of all UCs ($n = 215,985$) contained large predicted open reading frames (at least 100 amino acids long) that were predicted using the standard genetic code. Using alternative genetic codes may expand this set further by revealing novel, potentially different open reading frames generated from the UCs. A small proportion of these open reading frames contained domain signatures confirming the presence of currently unidentified organisms. Moreover, a comprehensive clustering analysis has led to the identification of UCs that were present across different human microbiomes (as well as from different samples/studies investigating the same human microbiome), indicating that we have discovered potentially widespread and as-yet-unclassified novel biological organisms within the human microbiome. The multimicrobiome clustering approach applied here provides an interesting way to understand the diversity and the distribution of the UCs across different microbiomes and geographical sites. For example, this approach led to the identification of 30 clusters that spanned 4 distinct microbiomes. The largest multimicrobiome cluster comprised 57 UCs recovered from saliva, sputum, oral, and lung microbiomes and was assembled from 12 different samples. Although it is impossible to identify the true clusters present in the data due to the novelty of the UCs, the clustering approach helps to identify obvious patterns of sequence similarity between microbiomes and studies. This approach provides an additional dimension by capturing unknown sequences that are shared between different projects or human microbiomes.

Virus predictions carried out by DeepVirFinder—a machine-learning-based virus prediction tool for identifying viruses from metagenomic data sets—have shown that approximately 50% of all UCs are likely to be of virus origin. Additionally, nearly 30% of all UCs identified in this study have an overlap with uncultivated viral genomes currently catalogued in IMG/VR databases. As with most similarity-based approaches, we used an arbitrary threshold for determining a match to the IMG/VR database, and thus, a match does not mean the sequences are closely related. Interestingly, this study provides an added dimension to these matching uncultivated viral genomes (UViGs) by providing information on the type of microbiome in which they have been found. It is anticipated that UCs catalogued in this study may have some overlap with other viral genome databases such as the Gut Phage Database (51) and Gut Virome Database (52). Short contigs, i.e., those less than 1 to 5 kb, are often ignored in most data mining, and exploration research typically in studies that employ a contig binning step as binning has been shown to be less sensitive for short contigs (50, 53, 54). The clustering and time point analyses carried out on short UCs have shown that these short UCs are originating from biological entities and predominantly represent the novel microbial sequences that are currently uncatalogued. This has been demonstrated with the example of short circular sequences with terminal repeats. Short contigs are typically

excluded from large microbiome mining studies employing the metagenomic binning approach but were studied in detail here. These short UCs are found across multiple human microbiomes and samples; we speculate that these are of viral origin and could potentially represent novel CRESS DNA or satellite viruses, although the ORFs originating from these genomes do not bear any sequence of functional similarity to the typical rep and cap genes. Moreover, a number of large contigs were found to contain various functional ORFs and domains often originating from virus or phages, indicating that a proportion of UCs are very likely to be novel viruses that infect currently uncharacterized microbes. In our approach, we have implemented a protein sequence similarity-based identification that enables the identification of distantly related sequence homologues (49). This approach can potentially "classify" contigs of viruses or phages as their corresponding host with very low sequence similarity. Indeed, viruses are well known to mimic their host genomic signatures by incorporating genomic sequences from their host into their genome. We anticipate that the virus diversity described in this work is reasonably underestimated due to this specific characteristic of viruses and speculate that a range of assembled contigs classified as bacterial with very low sequence similarity across a short genomic coverage is likely to be of virus origin. This hypothesis will need to be tested further by mining the "known" and "partially known" contigs systematically. We note that a range of UCs matching known and partially known sequences could be taxonomically uncharacterized in GenBank databases such as unclassified viruses. Assembled contigs matching these sequences are categorized as known (protein sequence similarity >80%) or partially known (protein sequence similarity <80%) in this study. Those contigs would need to be investigated further to identify potentially novel and divergent sequences assembled in this study. The HMP control sample analyses resulted in only a few UCs, validating the UC identification approach implemented in our framework. The results generated from this study can be extended to identify the organisms that cooccur in different microbiomes, which in turn can help to inform the interactions between these organisms and how they affect their hosts—humans. Despite our having sequenced human microbiomes extensively, our understanding about how these microbes interact with humans remains limited. These large-scale explorations can help to understand the human holobionts and the interactions of macro- and microorganisms. Based on these results, we do not know whether the microbes identified in different studies are consistently associated with humans or whether they are just passing associations captured at the time of sampling; the latter would make it even harder to make comparisons between samples and microbiomes.

The UC landscape changes over time as more sequences get characterized and added to the ever-expanding sequence repositories. This was demonstrated by comparing the UCs to different GenBank databases over the course of 18 months. We have estimated that 1.64% of the UCs identified in this study are getting characterized each month. However, this number would be highly dependent on the types of data deposited in the International Nucleotide Sequence Database Collaboration (INSDC) resources. This study provides a strong foundation of preliminary estimation of this rate, and UCs would need to be analyzed at multiple future time points to determine how the rate at which the UCs are being classified changes over time. Additionally, the time point analysis also provides strong evidence of the real biological entities being assembled and characterized in our study. Indeed, a proportion of the UCs was taxonomically classified during the period of the study. This delineation of the UCs demonstrates that the unknown matter that surrounds us largely belongs to currently uncultured, unidentified microbes that we interact with on a daily basis. The technological advances have accelerated the speed at which genomic sequences belonging to novel uncultured organisms are being deposited in INSDC databases. This sharp increase of metagenomically assembled microbial genomes has led to the scientific community driving the development of genomic data and metadata standards such as MIMAG (for bacteria and archaea) (55) and MIUVIG (for viruses) (47) for consistency and comparison purposes. The taxonomic classification landscape has also faced a tectonic shift

whereby it is moving from phenotype-based classification to more holistic sequence-centric phylogenetic classification, e.g., GTDB (bacteria and archaea) (56) and ICTV (viruses) (57). These changes enable the incorporation of the uncultured sequence diversity into the microbial taxonomy and will provide a more comprehensive understanding of the complex phylogenetic relationships and interactions between different microbes.

The metagenomics analysis framework developed here works as a proof of concept for overcoming the challenge of the quantification of the unknown in already "analyzed" data sets. The pipeline developed here is flexible and can be applied to any microbiome. To get a cross section of different human microbiomes and geographical locations while keeping the overall data set size manageable, large studies involving >100 samples were discounted. This framework can readily be applied to routine metagenomic exploration, which can help to gain further understanding of the landscape of sequences of unknown origins. However, the framework applied here is easily portable to metatranscriptomics data. In fact, a couple of the BioProjects (PRJEB10919 and PRJEB21446) analyzed in this study were indeed from a metatranscriptomic study. It is important to note that, unlike other studies that often focus on the cross-assembly of different samples, each sample was assembled individually here. This is regarded as best practice when a cocktail of samples from unrelated studies is analyzed in bulk. The coassembly would often lead to fragmented assembly as the complexity of sequences originating from multiple samples would be much higher than for a single sample (58). In contrast, independent assembly is expected to capture better diversity across each sample with high-quality genomes assembled from each sample (58). Typically, the sequence similarity-based approach is less reliable for unrelated sequences as the similarity search tools heavily rely on the databases used in the analysis. Like most other pipelines, this framework classifies the sequences with respect to a static version of the reference sequence databases. The search results are as good as the data in the ever-expanding repositories that are often too large to be hosted on a local computer. In order to improve this, an alignment-free approach could be explored. The development of a general-purpose alignment-free prediction method that can categorize the sequences based on the genomic composition would be suitable for the downstream analysis of the UCs. The UC classification is highly dependent on the methods employed to identify and quantify the unknown. Moving away from the sequence similarity-based methods would help to categorize and classify the currently unknown sequences better. Machine-learning-based approaches might be deemed suitable in certain circumstances to overcome the similarity threshold-based approaches. In the case of completely novel sequences that bear no similarity to currently known sequences, significantly rigorous training sets and features would need to be identified and be built into the models in order to make accurate predictions, as machine learning approaches are highly reliant on the training data with which the models have been developed. Moreover, a recent study by Krishnamurthy and Wang (59) made predictions for picobirnaviruses to be bacteriophages rather than eukaryotic viruses based on the presence of bacterial ribosome-binding sites in front of the coding sequences. This approach could potentially be applied to check whether viral UCs are bacteriophages.

**Conclusion.** This study demonstrates that there is a large diversity of unknown sequences embedded within various human meta-omic samples available in public repositories. It is clear that the unknown sequence landscape observed in this study is likely to be the tip of the iceberg, and as we scan more microbiomes and extend this to less-studied environments, e.g., insect metagenomes, we are likely to gather a better understanding of the unknown sequence space. As more species and environments are sequenced more readily, the rate at which the unknown sequences become known would also change. Our results of novel viruses indicate that the unknown microbes and their genomic signatures are likely to be more divergent from those currently present in widely used sequence databases; however, it should be noted that many of the short contigs found in our study are likely to represent fragments of larger viral genomes rather than being short but complete viral genomes. Our study also shows

that at least some of these unknown microorganisms are prevalent in nature. To overcome this, more comprehensive resources including searchable databases such as those enabled using BIGSI (60) and federated indexes (61) could be created for the unknown sequence data and metadata. This would allow researchers to explore the human metagenomic sequence space in a more holistic manner and, in turn, provide a better understanding of microbial diversity interacting with and within human hosts. It would enable researchers to search, link, and explore the unknown sequences present in different microbiomes, studies, and samples. Such resources could help in speeding up the pace at which unknown sequences can be "classified" and make it easier for researchers to determine the functional and/or ecological importance of the organisms from which the sequence comes. A concerted effort could help to pin down human-microbial interactions in a broader context such as linking unknown microbes to human diseases and disorders of unknown etiologies.

## MATERIALS AND METHODS

This study includes the data sets available within the EBI MGnify resource. All human microbiome studies submitted to ENA which were included in the MGnify databases were downloaded with the corresponding metadata on 19 April 2019. In order to obtain detailed metadata, each study was linked to the corresponding SRA repository using NCBI E-utilities (62). As the focus was on shotgun metagenomic data sets, studies targeting metabarcoding-based sequencing methods such as 16S and amplicon sequencing were excluded as well as studies that solely focused on third-party annotation, i.e., analysis of previously published data, and lacked primary data. In order to reduce sequencing technology-related bias, the studies that utilized sequencing platforms other than Illumina were excluded. Very large studies involving >100 samples were discounted in order to get a cross section of different human microbiomes and geographical locations while keeping the overall data set size manageable. The filtered set initially comprised 44 distinct studies with 1,130 samples of which 1,121 were available to download. A script that uses parallel-fastq-dump (63) was developed to download reads in fastq format. In total, 1,121 samples (789 paired-end [PE], 332 single-end [SE]) from 43 distinct studies were successfully downloaded and submitted to the pipeline. Out of 1,121 samples, 158 could not be assembled due to insufficient reads and were excluded from downstream analysis (see Text S1 in the supplemental material). In summary, 963 (784 PE, 179 SE) samples from 40 distinct studies were included and were processed using the complete metagenomic analyses pipeline described below (Fig. 2).

This study set included a range of different sample types as described in Fig. 3. It is important to note that this set is highly skewed toward the human gut metagenome that is normally sampled through fecal material, and the oral microbiome was the second most common sample type included in the study. Although other metagenomes were underrepresented, our study covered a wide range of samples from various human body sites and bodily fluids. A miscellaneous metagenome labeled only as "Human" was included in this data set that represents 3 distinct studies including PRJEB14301 (cerebrospinal fluid [CSF], $n = 1$), PRJEB21827 (A/B testing for colon model, $n = 12$), and PRJEB6045 (metagenomics of medieval human remains from Sardinia, $n = 1$).

In order to assess the quality of the samples and remove sequencing adaptors, all samples were processed through BBDuk from BBTools package (64). BBDuk autodetected the presence of the relevant adaptor sequences from the input files specified and trimmed them. Additionally, commonly known sequencing contamination and spike-in sequences were also removed as part of this quality control (QC) step. All reads that passed QC were retained and mapped to the human genome sequence build GRCh38 using the Burrows-Wheeler aligner (BWA) (65), and unmapped reads were subsequently extracted using SAMtools (66). BBNorm (64) was used to normalize reads based on the kmer coverage composition with a kmer threshold of 3 (mindepth = 3). This step also enabled the acceleration of the assembly process as only a subset of reads was used to build the *de novo* assembly and resulted in better assembly quality overall (67). The read lengths varied widely between the samples and the studies; thus, it was not possible to compare the quality metrics using the read-based measures as it would be misleading. To enable a comparison, quality assessment metrics were carried out for a number of bases.

**De novo assembly and taxonomy label assignment.** The normalized reads were *de novo* assembled using the SPAdes (68) assembly pipeline, with the default parameters. A script was developed to extract contigs that were longer than 300 bases as short contigs do not contain a lot of information and they were excluded from downstream analysis as a precautionary measure. Although the normalized subset of reads was used to generate assemblies, these reads cannot be used to assess the assembly quality as they represent a small subset of the actual reads. To assess the assembly quality, the complete set of reads that did not map to the human genome was mapped onto the *de novo*-assembled contigs with BWA (65) using the default parameters. The assembly quality statistics such as coverage, length, and number of mapped reads were generated for each contig using pileup.sh from the BBTools package (64).

Contigs were searched against the GenBank nonredundant (nr) protein databases using the BLASTX algorithm implemented in DIAMOND (69). It carries out a six-frame translation of the nucleotide sequences and then searches those translated sequences against the nr protein databases. This step enables the identification of distantly related homologues of the currently known sequences. The default DIAMOND tabular output format with additional columns 'qframe, staxids, stitle' was generated for

aligned sequences. The top 25 hits for each contig were extracted and analyzed downstream (−E value = 0.001). The lowest common ancestor (LCA) of these hits was computed, and superkingdom was assigned based on the LCA using the Python ete3 package (70). The contigs that did not have any protein match were extracted and searched against the GenBank comprehensive nucleotide database (nt) using BLASTN (−E value = 0.001); BLAST output format 7 with additional columns 'qframe, staxids, stitle' was generated. This step helped to identify and remove noncoding sequences such as rRNA and untranslated regions of currently sequenced organisms included in the databases.

To identify the geographical distribution of the raw data, location data were mined from the SRA metadata resources using pysradb (71) for each study. Geolocation information was available for 861 samples as shown in Fig. 4. A complete list of study location is shown in Table S1 in the supplemental material. These samples were sequenced in various sequencing facilities across the world, and the complete distribution of the sequencing centers is shown in Fig. S6a.

**Unassembled sequences.** "Unassembled" bases are defined as bases from reads that did not map to the human genome and could not be assembled into contigs. These were calculated from reads that did not map to assembled contigs. These bases/reads could not be classified as part of this project but were quantified as shown in Fig. S6c (gray bars). Our quantification suggests that almost all microbiome samples have a proportion of unassembled sequences, and on the sample, the average value for this is around 23.91% (standard deviation [std], 26.59%). This unassembled sequence proportion was very high for samples originating from PRJEB15334 (mean, 51.17%; maximum, 97.67%; std, 24.30%) and PRJEB17784 (mean, 82.59%; maximum, 98.83%; std, 17.84%). Overall, 8.18% of all data fell into this category as described in Fig. S6b. A range of possibilities from degraded nucleic acid to sequencing protocols could lead to poor-quality data that cannot be used for *de novo* assembly.

**Control samples.** The Human Microbiome Project mock community samples ($n$ = 9) were downloaded for study PRJNA298489 and were analyzed using the metagenomic framework described above for quality control and workflow assessment. This would also allow us to validate the metagenomic analyses pipeline for this study.

**Postmetagenomic analysis.** All unknown contigs (UCs) were analyzed further to get insights into the coding potential of those sequences. The getorf tool from the EMBOSS (72) suite was used to generate open reading frames (ORFs) from contigs (-find 1, -minsize 300) using the standard genetic code. These ORFs were searched against a range of different domains and functional identification databases included in the InterProScan.

To explore the sequence similarity between samples and the diversity of the unknown sequences, a nucleotide-based sequence similarity clustering which also used coverage was carried out using MMSeqs2 (45, 73). All sequences with at least 90% sequence identity and at least 80% overlap were clustered using the MMSeqs2 easy-cluster pipeline (45). All UCs were processed through the CheckV (48) pipeline to identify the UCs that were likely to belong to viruses.

The most widely applied sequence similarity-based approaches rely on static versions of the databases to carry out the classification step of the analysis. In this study, the sequence databases utilized were downloaded on 18 April 2019. All results included in the study are based on the searches against this static version of the databases. However, the sequence database is ever-expanding with new sequences being added to the databases each day. With newer sequences being added to these databases, it is very likely that unknown sequences transition into the "known sequence space" over time. In order to identify the proportion of the unknown sequences classified over the period of the study, 4 distinct time points were considered. Static versions of the databases were downloaded on 31 October 2019, 5 March 2020, and 14 October 2020.

To predict the proportion of UCs that are likely to be viruses, the virus prediction tool DeepVirFinder was used. DeepVirFinder has been demonstrated to accurately predict viruses from metagenomic data sets and has been shown to work well even with short contigs (74). It was deemed suitable for UCs as a large proportion of UCs identified in this study are under 1 kb long. DeepVirFinder was run on all UCs with default parameters, and $q$ values (false-discovery rate) were computed for the predictions using the R library $q$ value as recommended in the DeepVirFinder tutorial. The $q$ value output was rounded to 3 decimal points, and a cutoff $q$ value of <0.05 was applied.

In order to identify if the UCs captured in this study have any overlap with other uncultured virus databases such as IMG/VR (36), initial nucleotide (BLASTN) and protein sequence-based (BLASTX and BLASTP in DIAMOND) searches were carried out against nucleotide and protein sequence data downloaded for the latest IMG/VR version, 2020-10-12_5.1. BLASTN searches were carried out with default parameters, except for the E value, which was set to 0.0001, and the output was generated in standard tabular format. For BLASTP searches, predicted ORFs were used.

**Data availability.** All assembled unknown sequences generated here were submitted to ENA as third-party annotations and are accessible through BioProject PRJEB41812. Results and code generated in this study are available on Zenodo (https://zenodo.org/record/5907223) and GitHub (https://github.com/sejmodha/UnXplore).

## SUPPLEMENTAL MATERIAL

Supplemental material is available online only.

**TEXT S1**, PDF file, 0.1 MB.

**FIG S1**, EPS file, 0.2 MB.

**FIG S2**, EPS file, 4.5 MB.
**FIG S3**, EPS file, 2.8 MB.
**FIG S4**, EPS file, 0.3 MB.
**FIG S5**, EPS file, 0.8 MB.
**FIG S6**, EPS file, 2.0 MB.
**FIG S7**, EPS file, 0.3 MB.
**TABLE S1**, CSV file, 0.005 MB.
**TABLE S2**, CSV file, 0.001 MB.

## ACKNOWLEDGMENTS

We thank Quan Gu, Maha Maabar, Sreenu Vattipally, Josh Singer, Hilary Fawcett, and Andrew Davison for contributing to an internal hackathon organized in 2016 where the idea of mining metagenomic data sets for identification of "dark matter" was developed.

S.M. is funded by an MRC Precision Medicine PhD studentship (MR/S502479/1). D.L.R., J.H., and R.J.O. are funded by the MRC (MC_UU_1201412).

We declare that we have no competing interests.

Conceptualization, S.M., J.H., R.J.O. Data curation, formal analysis, project administration, investigation, resources, software, validation, visualization and writing—original draft, S.M. Funding acquisition, S.M., D.L.R., J.H., R.J.O. Methodology, S.M., J.H., R.J.O. Supervision, D.L.R., J.H., R.J.O. Writing—review & editing, S.M., J.H., R.J.O.

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
