## [Reviewer comments · mSystems]

Quantifying and cataloguing unknown sequences within human microbiomes

Sejal Modha, David Robertson, Joseph Hughes, and Richard Orton

Corresponding Author(s): Sejal Modha, MRC-University of Glasgow Centre For Virus Research

Review Timeline:

Submission Date:	December 14, 2021
Editorial Decision:	January 26, 2022
Revision Received:	February 9, 2022
Accepted:	February 9, 2022

Editor: Mani Arumugam

Reviewer(s): The reviewers have opted to remain anonymous.

Transaction Report:

DOI: <https://doi.org/10.1128/msystems.01468-21>

January 26, 2022

Dr. Sejal Modha
MRC-University of Glasgow Centre For Virus Research
Sir Michael Stoker Building
464 Bearsden Road
Glasgow G61 1QH
United Kingdom

Re: mSystems01468-21 (Quantifying and cataloguing unknown sequences within human microbiomes)

Dear Dr. Sejal Modha:

Thank you for submitting your manuscript to mSystems. We have completed our review and I am pleased to inform you that, in principle, we expect to accept it for publication in mSystems. However, acceptance will not be final until you have adequately addressed the reviewer comments.

Both reviewers are supportive of the main message of the manuscript. The remaining criticism is on acknowledging the limitations of the approach used here. Please revise your manuscript with this in mind.

Preparing Revision Guidelines

Sincerely,

Mani Arumugam

Editor, mSystems

Journals Department
Reviewer comments:

Reviewer #2 (Comments for the Author):

The manuscript from Modha et al. responds to all of the points raised by the reviewers after the initial submission. I appreciate the effort of the authors to confirm the results with DeepVirFinder, and I believe the additional analyses strengthen the message of the manuscript. Overall, the message conveyed about the need of looking at unclassifiable contigs is important and should be heard. I encourage a few more minor corrections:

1. DeepVirFinder classifies many of the short contigs as viral, but does not prove that those are complete (or near-complete) viruses. Moreover, at line 406, authors write that short UCs often cluster with (partially) known contigs. Some short UCs may be fragments of known viral entities. While no further analysis is required here, authors should underline that short contigs are likely "fragments" of viruses, rather than viruses or viral genomes. This for example should be specified at lines 397~402, and in the conclusions, as a potential limitation of the study.
2. Please indicate more clearly that viral databases other than IMG/VR are available, and that there is possible overlap with those as pointed out by Reviewer 1 (point 3).
3. Lines 418-419: please indicate percentages after absolute values as done at previous lines
4. Figure 4: please enlarge the font size of pie-charts and in-country-counts to enhance readability
5. Figure 6: Does "number of microbiomes" refer to the number of distinct "microbiome types" that contribute to each cluster (i.e. gut, oral, skin etc...)? The caption is not very clear and should explain the stratification (i.e. the four different colors) better. If this is the case, consider using a more explicit "number of body districts = ..." instead.
6. Please add a supplementary table (or a separate file on the linked Zenodo repository) with each UCs and its DeepVirFinder results, q-values, CheckV score and other relevant data for classification.

February 9, 2022

Dr. Sejal Modha
MRC-University of Glasgow Centre For Virus Research
Sir Michael Stoker Building
464 Bearsden Road
Glasgow G61 1QH
United Kingdom

Re: mSystems01468-21R1 (Quantifying and cataloguing unknown sequences within human microbiomes)

Dear Dr. Sejal Modha:

Your manuscript has been accepted, and I am forwarding it to the ASM Journals Department for publication. For your reference, ASM Journals' address is given below. Before it can be scheduled for publication, your manuscript will be checked by the mSystems production staff to make sure that all elements meet the technical requirements for publication. They will contact you if anything needs to be revised before copyediting and production can begin. Otherwise, you will be notified when your proofs are ready to be viewed.

Publication Fees:

We recognize that the video files can become quite large, and so to avoid quality loss ASM suggests sending the video file via <https://www.wetransfer.com/>. When you have a final version of the video and the still ready to share, please send it to mSystems staff at mssystemsjournal@msubmit.net.

For mSystems research articles, if you would like to submit an image for consideration as the Featured Image for an issue, please contact mSystems staff at mssystemsjournal@msubmit.net.

Sincerely,

Mani Arumugam

Editor, mSystems

Journals Department
Figure S2a and Figure S2b: Accept
Figure S5a, Figure S5b, Figure S5c: Accept
Figure S1: Accept
Table S1: Accept
Supplementary text: Accept
Table S2: Accept
Figure S3: Accept
Figure S7a and Figure S7b: Accept
Figure S6a, Figure S6b, Figure S6c: Accept
Figure S4: Accept